# Do all roads lead to Rome? An ideal-type study on trajectories of resilience in advanced cancer caregiving

Sophie Opsomer[1,2]*, Luca De Clercq[3], Jan De Lepeleire[1], Sofie Joossens[4], Patrick Luyten[3,5], Peter Pype[2,6], Emelien Lauwerier[2,7,8]

1 Academic Centre for General Practice, Department of Public Health and Primary Care, KU Leuven, Leuven, Belgium, 2 Department of Public Health and Primary Care, Ghent University, Ghent, Belgium, 3 Clinical Psychology, Faculty of Psychology and Educational Sciences, KU Leuven, Leuven, Belgium, 4 Program of Health, University Colleges Leuven - Limburg, Leuven, Belgium, 5 Research Department of Clinical, Educational and Health Psychology, University College London, London, United Kingdom, 6 End-of-Life Care Research Group, Ghent University campus, Ghent, Belgium, 7 Department of Experimental Clinical and Health Psychology, Ghent University, Ghent, Belgium, 8 Department of Health Psychology, Faculty of Psychology, Open University, Heerlen, Netherlands

* sophie.opsomer@kuleuven.be

**Data Availability Statement:** This paper employs a 'minimal data set' (consisting of narratives and illustrating quotes) in order to highlight the conclusions drawn. However, the interviews

## Abstract

### Objective

Studies on resilience in advanced cancer caregiving typically focus on the interplay between resilience-promoting resources and coping strategies that may be associated with resilience. However, no studies have investigated the emergence of trajectories of resilience and distress in individuals confronted with a cancer diagnosis of a loved one.

### Methods

Ideal-type analysis, a method for constructing typologies from qualitative data, was used to identify trajectories involving resilience or the lack thereof based on fifty-four interviews conducted with seventeen partners of patients recently diagnosed with advanced cancer over a period of three years.

### Findings

Six trajectories could be distinguished, three of which involved resilience (rapidly adapting resilience, gradually adapting resilience, and slowly adapting resilience), while the other three trajectories (continuing distress, delayed distress, and frozen disconnection) reflected a less optimal adjustment. These different trajectories seemed to be rooted in the individual characteristics of partners, the behavior of a support network, and interactions between the two.

### Conclusion

The differentiation between these trajectories in partners of patients diagnosed with cancer not only furthers research on resilience in the face of adversity, but also promises to assist

contain potentially highly sensitive information and involve indirect identifiers. For privacy and confidentiality reasons, and mandated by the Ethical Commission of University Hospitals Leuven, the interviews and/or the interviewer's field notes cannot be published or distributed. The complete datasets used and analyzed during the current study are deposited in the repository of the Academic Center of General Practice (ACHG), KU Leuven and are available from the corresponding author or from the head of the department, Prof. Dr. Birgitte Schoenmakers, Kapucijnenvoer 7 / 7001, 3000 Leuven (birgitte.schoenmakers@kuleuven.be) on reasonable request, and after the inquirer can provide written permission by the Ethics Committee Research UZ / KU Leuven, Herestraat 49, 3000 Leuven (ec@uzleuven.be). The interview transcripts will be delivered in the original language (Dutch) and will be anonymized and de-identified.

**Funding:** The author(s) received no specific funding for this work.

**Competing interests:** The authors have declared that no competing interests exist.

healthcare professionals in optimizing support for this often-neglected group of partners of patients diagnosed with cancer.

## Background

Receiving the diagnosis of advanced cancer may have a tremendous impact not only on the patients themselves but also on their partners [1]. Moreover, the impact of such a diagnosis may have long-lasting effects well after the patient's death [1, 2]. In fact, a loved one being diagnosed with advanced cancer can be considered a potentially traumatic event (PTE) and consequently can put the partner at risk of developing anxiety, depression, or post-traumatic stress disorder (PTSD) [3]. However, most caregivers seem to adapt well to the diagnosis of advanced cancer and manage to return to a stable status of mental wellbeing [4]. Furthermore, in the aftermath of PTEs, several outcome trajectories (resilient, recovered, delayed, and chronic) have been identified [5, 6]. As such, in the minimal impact resilience trajectory (the most common trajectory following a PTE), symptoms of distress are limited to a brief period immediately following the PTE, after which follows a period characterized by few or no symptoms of distress and the ability to function healthily [7]. Recovery means that symptoms of moderate to severe distress abate gradually over time, eventually resulting in a baseline level of functioning. In a chronic distress trajectory, severe distress continues unabated. Delayed reactions involve the absence of distress or subthreshold symptoms levels that worsen over time. Nevertheless, a growing body of evidence demonstrates that, following a PTE, the resilience trajectory is very typical (60–90%) since the majority of people experience a relatively short episode of mental distress succeeded by a stable trajectory of healthy functioning across time and resulting in a resilient outcome, such as re-established mental wellbeing or personal growth [5, 8–10]. To the best of our knowledge, there are no quantitative studies available on the prevalence of resilience in family caregivers of loved ones diagnosed with advanced cancer. Nevertheless, based on the authors' clinical family practice experience, it is reasonable to assume that most partners follow a resilience trajectory and will have a resilient outcome. Furthermore, several recent studies on resilience in advanced cancer caregiving report on the interplay of resilience-promoting resources, including individual characteristics (flexibility, positivity, inner strength, ability to control the flow of information, and asking for and accepting help), as well as the availability of a support network and the resulting resilience outcome [4, 11, 12]. One study even investigated what happens to the resilience-promoting resources and ensuing coping strategies when intimate partners of cancer patients are confronted with two PTEs happening concurrently [13]. However, to the best of our knowledge, no studies investigated the trajectories of resilience and distress elicited by a family member's diagnosis of advanced cancer over time. As a result, insight into these trajectories in partners of patients with advanced cancer, and into how and when the resilience-promoting resources are applied, is lacking.

To fully capture the complexity of factors influencing psychosocial adjustment while caring for a loved one diagnosed with advanced cancer, a qualitative study design seemed preferable [14]. Indeed, by starting with the participants' lived experiences, more insight into a complex phenomenon such as resilience can be generated. Additionally, a qualitative approach can outline a broader understanding of influencing contextual factors. In this context, an ideal-type analysis–a methodology that seeks to identify groupings of participants who share similar experiences–offers a systematic and rigorous method for constructing typologies of trajectories [14].

This study therefore aims to identify possible differences in resilience in partners of patients with an advanced cancer diagnosis over a period from one to three years following the patient's diagnosis of advanced cancer. The research questions are: 1) What different types of resilience trajectories can be distinguished in partners of patients diagnosed with advanced cancer? 2) How are resilience-promoting resources involved in the development of these trajectories?

## Methodology

### Study design

A longitudinal qualitative study was conducted with 17 partners of patients diagnosed with advanced cancer. To identify trajectories of resilience, or lack thereof, ideal type analysis was used to analyze data from 54 interviews conducted over a three-year period.

### Participants and procedures

The study was advertised via flyers in the waiting rooms of the oncology wards of the University Hospitals of Leuven (from 01/11/2020 to 01/02/2021) and Ghent (from 01/11/2020 to 01/02/2021) and the Imelda Hospital in Bonheiden (from 15/06/2020 to 01/02/2021). The closure of the hospitals due to the COVID-19 pandemic necessitated the extension of recruitment to general practices and peer group websites of oncology patients (from 15/07/2020 to 19/02/2021). Nineteen candidates contacted the researcher by e-mail or telephone.

Inclusion criteria:

- Being the partner and principal caregiver of a person recently (less than one year prior) diagnosed with cancer in an advanced or palliative stage. Advanced stage cancer is defined as cancer in stage III, IV, or metastatic cancer. Cancer in a palliative stage means that a cure is no longer achievable or one's life expectancy is one year or less.

- Adults under 65 years of age.

- Fluency in Dutch.

Exclusion criteria:

- Partners with diagnosed depression or psychological illness before the cancer diagnosis.

- Partners of patients with a life expectancy of three months or less.

### Data collection

Nineteen partners applied to participate in the study. One candidate did not meet the inclusion criteria, and one dropped out after the first interview as she no longer wished to discuss the cancer. Seventeen candidates were included in the study. They were invited for an interview every six months from inclusion until death of the patient. The closing interview was conducted approximately six months after the patient's death (seven interviews) or upon completion of the study in November 2022, in the case the patient was still alive (eight interviews). One participant dropped out after three interviews without further clarification. Furthermore, one patient died a few weeks before completion of the study, hence a closing interview with the partner during the study period was no longer feasible. Fifty-four semi-structured interviews were conducted based on an interview guide (provided as S1 File) and were transcribed verbatim. All interviews were conducted in Dutch. Due to COVID-19 measures, most interviews took place via Zoom and were video recorded. Among the others, two candidates

preferred being interviewed at home; five interviews were conducted at the researcher's practice and were audio recorded; and two interviews were conducted in writing per the participant's request. Apart from five participants who preferred the patient be present for at least one interview, the patients did not take part in the interviews.

The Dutch Mental Health Continuum-Short Form (MHC-SF), a validated fourteen-point questionnaire that assesses emotional, social, and psychological wellbeing [15] was sent to the participants every two months. Fifteen participants returned the questionnaire on at least two occasions.

## Methodological framework

To explore the trajectories of resilience in cancer caregiving, an ideal-type analysis was chosen since it constructs typologies from qualitative data by systematically and rigorously comparing cases within a dataset to form 'ideal types' or groupings of similar cases [14]. The ideal types can be considered constructed generalizations of a phenomenon without the intention of representing reality. However, by organizing the data into ideal types novel insights in reality can be gathered [14].

To investigate in-case and across-case patterns in trajectory-influencing features and the effect of resilience-promoting resources, Saldana's 'longitudinal qualitative data summary matrix' was employed as suggested for longitudinal qualitative studies [16]. As directed, the matrix was completed for each participant and at each intervening event with data regarding resilience-promoting resources and coping strategies used. In this way–per trajectory and across trajectories–changes and evolutions in the resilience-promoting resources used and the resulting coping strategies could be mapped and compared.

### Data analysis

The ideal-type analysis was carried out in eight steps [14]:

1. Familiarization with the dataset by watching and listening to the video and audio recordings and by reading and re-reading the transcripts.

2. Writing a narrative reconstruction of the participants' stories focusing on the study aims.

3. Constructing the ideal types by exploring the similarities and differences among cases to identify patterns across the dataset. In order to identify patterns in the evolution of distress during the follow-up period, the participants' level of distress following each event was independently assessed by three different researchers making use of an assessment form specifically designed for that purpose. The assessment form was discussed within the research team and adapted until consensus was reached about the criteria for each level of distress (mild, moderate, severe). The assessment form is provided as S2 File. All three researchers constructed the participants' trajectories of distress independently. Subsequently, the individual trajectories were discussed until consensus was achieved. In the next step, the individual trajectories were compared, and the ideal-type trajectories were identified.

4. Identifying the 'optimal case' to represent each ideal type.

5. Naming and describing the ideal types.

6. Checking the credibility of the ideal types by a researcher not involved in the analysis thus far.

7. Mapping the influencing factors by using Saldana's 'longitudinal qualitative data summary matrix' [16].

8. Writing up the study findings and translating the illustrative quotes from Dutch to English.

An overview and description of the steps and the authors involved is provided in Fig 1.

## Trustworthiness

The quality of qualitative studies is judged by their trustworthiness which includes several criteria, such as credibility, transferability, reliability, and reflexivity [17, 18]. In a longitudinal qualitative study trustworthiness can be challenged by researcher bias as the interviewer and the participants inevitably get to know each other through repeated interviews. This relationship may affect the depth and quality of the data obtained [19]. In addition, credibility may be threatened because researchers may have preconceived notions that affect how they interpret

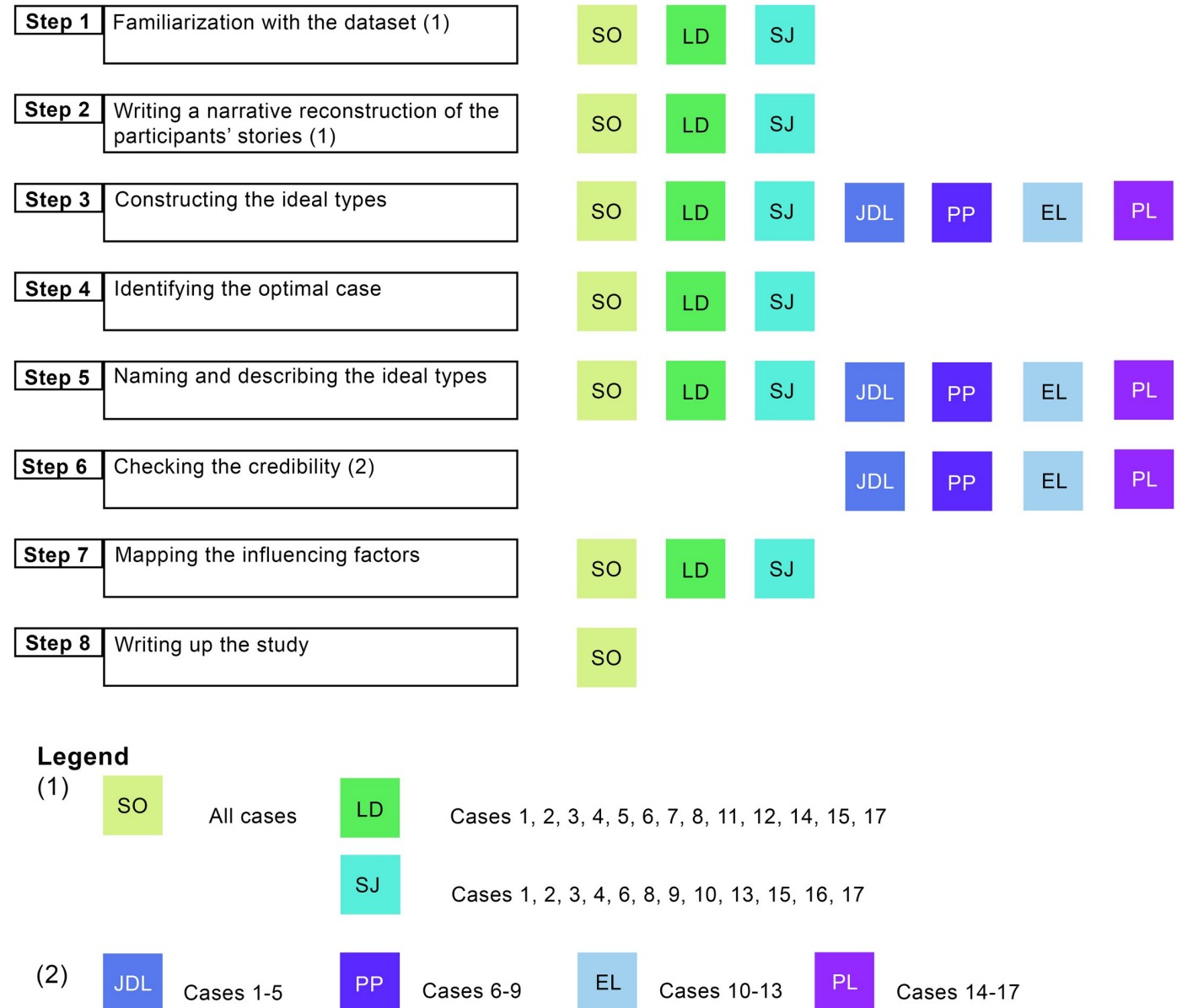

**Fig 1. Ideal type analysis: Overview and authors involved.**

data and draw conclusions [19]. However, to ensure the trustworthiness of the study by increasing reflexivity, the interviewer's field notes were considered. A debriefing with the participant and a short peer debriefing of the interviewer with the study supervisor following each interview was organized to increase the credibility and reliability. In addition, to optimize the relevance, credibility, and completeness of the data, the study protocol and interview guide were developed in consultation with someone who had experienced the loss of a partner to cancer. To further ensure trustworthiness, credibility, and transparency the interviews belonging to the first three cases were analyzed by three researchers independently. The preliminary findings were discussed between them and presented to the supervisors and co-authors before moving on to subsequent cases. The 14 remaining cases were divided among three researchers (SO, LDC, SJ). Each case was analyzed separately by two researchers. However, the corresponding ideal-type trajectory was identified by all three researchers. Subsequently, the narratives of the cases were divided among the research team members not involved in the analysis to this point (JDL, PL, PP, EL). They regrouped the cases into the proposed ideal-type trajectories. Disagreements and ambiguities about the description of the ideal-type trajectories or the grouping of the cases were discussed within the research team until consensus was reached. Finally, the practicality of the ideal-type trajectories was discussed with and verified by someone who had experienced the loss of a partner to cancer.

## Research team

The first author, a family physician experienced in palliative care and qualitative research, initiated the study and conducted the interviews as part of her PhD project. She had neither a professional nor personal relationship with the participants and did not meet them before the first interview. The multidisciplinary research team consisted of two professors in primary healthcare, two professors in clinical (health) psychology, one doctor in biomedical science–all of whom are experienced in qualitative research–and one master student in pedagogy.

## Ethics

The interviews were conducted according to the COVID-19 measures in force at that time.

This study adhered to ethical standards established by institutional ethics committees and the principles outlined in the 1964 Helsinki Declaration for research involving human participants. All participants took part voluntarily in the study. Ethical approval was provided by the Ethics Committee Research UZ / KU Leuven on October 4, 2019, study number S63166; by the Ethics Committee of Ghent University Hospital on October 17, 2019, study number BC-06066; and by the Ethics Committee of Imelda Hospital Bonheiden on June 9, 2020. An amendment to extend recruitment to general practices and peer group websites of oncology patients was approved by EC UZ/KU Leuven on July 7, 2020 and by EC Ghent University Hospital on July 14, 2020.

Prior to commencing the study, written informed consent was obtained from all individual participants and their partners. The consent included permission for publication of anonymized and translated quotes from the interviews. Although the written consent covered all interviews, oral consent was also obtained before each interview. Participants were informed, both in writing and orally, that they could withdraw from the study at any time without providing a reason.

Furthermore, participant data has been anonymized in order to preserve the scholarly meaning while protecting the privacy of the individuals involved.

## Findings

### Demographic data

Demographic data outline for each ideal type is represented in Table 1.

In the illustration of the ideal types, all the participants' names have been changed. However, excerpts are verbatim but anonymized.

### Ideal-type trajectories

Six prototypical or 'ideal' trajectories could be distinguished, three of which demonstrated substantial resilience, whilst the other three trajectories reflected less optimal adjustment. Fourteen out of 17 participants could be classified under the trajectories reflecting resilience, while the trajectories characterized by a less optimal adjustment were represented by only one case each (three cases in total).

The six ideal types are described below, alongside the optimal case from each trajectory, and a summary of all cases that meet the description of the trajectory. A graphical representation is provided as Fig 2. More detailed narratives, including contextual features and illustrating quotes, are provided as S3 File.

Most partners experienced a peak in mental distress in the terminal phase of the disease, mostly starting from the moment the patient's physical condition worsened. However, this peak in mental distress did not seem to hamper the emergence of a resilient outcome (mental wellbeing or personal growth) following the partner's death.

**Ideal-type 1: Rapidly adapting resilience.** The advanced cancer diagnosis evoked a temporary status of moderate to severe mental distress, subsiding relatively quickly and returning to a baseline level of functioning without any significant lingering distress. Yet, whenever there was a new threat (new metastases, hospitalization, non-working therapy, new scan, patient being in a terminal phase, etc.), there was a relatively short episode (some days at the most) of moderate or severe distress, however without significant impact on longer-term average distress levels.

**Table 1. Demographic data for each ideal type trajectory.**

| | participant | | | | partner | | | |
|---|---|---|---|---|---|---|---|---|
| | Mean age | Gender | Education | Employment | Mean age | Gender | Education | Cancer diagnosis |
| **Rapidly adapting resilience** | 55.5 | Male 3 Female 1 | Secondary 1 Higher 3 | Active 4 Unemployed 0 Retired 0 | 55.5 | Male 0 Female 4 | Secondary 1 Higher 3 | Brain tumor Breast cancer Lung cancer |
| **Gradually adapting resilience** | 59.8 | Male 3 Female 3 | Secondary 3 Higher 3 | Active 3 Unemployed 0 Retired 3 | 64.6 | Male 3 Female 3 | Secondary 3 Higher 3 | Breast cancer Colon cancer Lymphatic cancer Neuro-endocrine cancer Pancreatic cancer |
| **slowly adapting resilience** | 49.7 | Male 0 Female 4 | Secondary 1 Higher 3 | Active 4 Unemployed 0 Retired 0 | 50 | Male 4 Female 0 | Secondary 1 Higher 3 | Brain tumor Lymphatic cancer Lung cancer |
| **Distress (Continuing, Delayed, Frozen Disconnection)** | 61.6 | Male 1 Female 2 | Secondary 2 Higher 1 | Active 1 Unemployed 1 Retired 1 | 58 | Male 2 Female 1 | Secondary 3 Higher 0 | Bone marrow cancer Pancreatic cancer Throat cancer |

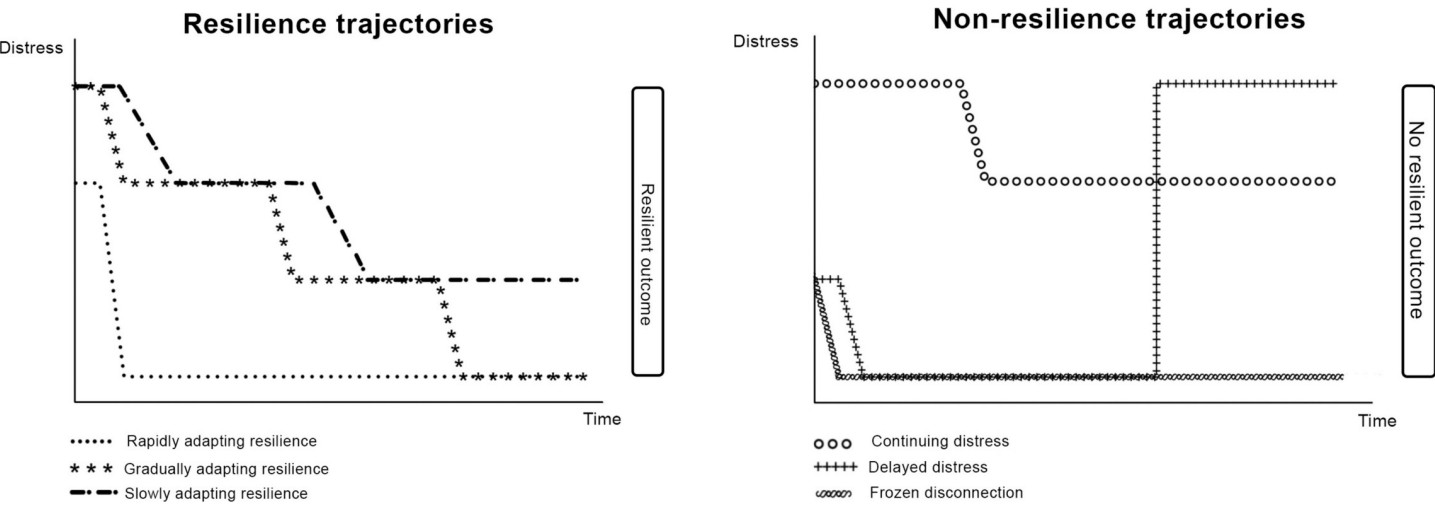

**Fig 2. Trajectories of resilience.** A graphical representation of six prototypical trajectories of adjustment to an intimate partner's diagnosis of advanced cancer: (A) Resilience trajectories; (B) Non-resilience trajectories.

*Optimal case*. Jim's story revealed a strong relationship with his wife Madelyn and his family. All family members were hardworking and did not show weakness nor did they communicate extensively about feelings. Madelyn's original cancer diagnosis, and some years later cancer in an advanced metastatic stage, caused moderate mental distress to Jim. However, driven by positivity and flexibility, Jim immediately sprang to action. He took up responsibility not only for his wife's care but also for his own wellbeing by breaking out of his so-called cocoon by meeting friends and volunteering.

"'My circle of friends enlarged [since Madelyn was diagnosed with advanced cancer]. That's because I never had any hobbies before and now I do [laughs]. On Sunday morning, I often go for a walk with a client who's become a good friend, and then we sometimes talk about my wife but mostly about other things. I really enjoy these walks. Furthermore, when my wife got sick, I also picked up volunteering at the youth movement. I now have a couple of good friends there, peers who are also involved [in the organization]. Two or three times a month, we try to do something fun. That's real friendship, you can feel that."

As such, Jim succeeded in mastering the situation by taking charge of household chores, engaging in new hobbies, learning new skills, such as cooking and nursing, reorganizing his work schedule to actively create intimate moments with his wife, and to be home whenever she needed him. His mental distress dropped and Jim soon found a new equilibrium in life without any significant mental distress, at least as related during the interviews.

"I don't believe that I've changed lately [as compared to the pre-diagnosis]. When Madelyn got that diagnosis, I was worried, but now, I guess that I'm functioning as before."

As Madelyn's prognosis worsened and physical decline became obvious, Jim's support network of family and friends became more important. The interactions within the support network intensified during the last days of Madelyn's life. Six months after her death, Jim was still visiting the cemetery several times per week to deal with his grief. Nevertheless, he talked about his enhanced self-confidence, his shift in priorities from working and making money to

enjoying life and nature, and in being aware of the importance of social involvement, all of which could be interpreted as signs of personal and post-traumatic growth.

> "Earlier, I would have said: 'I must work, think about the future, that's the most important'. But now, no, I don't really care about such things anymore. Why? Well, I don't think I've ever contributed anything to society before. But now I know that there are much more important things in life than just working and making money. I even took a leave. Something I'd not done in the last ten years. And I must say, I've immensely enjoyed it."

*Summary of other cases.* The participants from the rapidly adapting resilience trajectory–Jim, Rose, Lester, and Bruce–seemed to have in common a reliance on a sense of agency and inner strength, as well as positivity from the start. These individual resilience-promoting resources supported the participants in taking up responsibility for their partner and their own wellbeing and in mastering the situation relatively quickly, suggesting they successfully adapted their lives to the constraints caused by their partners' advanced cancer. Moreover, every upsurge of distress caused by a new event quickly abated by actively coping.

**Ideal-type 2: Gradually adapting resilience.**   The patient's diagnosis of cancer in the advanced stage was typically followed in these partners by a period of moderate to severe distress. Adaptation to the advanced cancer diagnosis and the multiple events surrounding this experience unfolded gradually and a variation in distress levels was apparent, progressively transitioning into a stable status of no distress. These partners also seemed to be able to better cope with subsequent challenges, such as news about the discovery of new metastases or hospitalizations of the patient.

The process was characterized by an intermediate personal growth. In addition, the participants' stories illustrated how they could find benefits in dealing with advanced cancer (e.g., a better relationship with the patient, feeling more appreciated or respected, etc.).

*Optimal case.* Three days after his father died of cancer, James' partner Kelly was diagnosed with a rare and atypical neuro-endocrine cancer, stage four. Urgent surgical intervention was needed. However, because of the COVID-19 measures, James was not allowed to enter the hospital. When Kelly was discharged the next day she felt very weak and fatigued, and neither the family physician nor the homecare nurse was allowed to visit with her. James was overwhelmed by feelings of grief, insecurity, anxiety, and loss of control. He repeatedly searched the internet for information about that type of cancer, and it irritated him that he could not find any facts or numbers about its prevalence or prognosis. However, when the oncologist admitted that the lack of statistical data and exact numbers about the cancer made her feel uncomfortable and concerned, James felt understood and empowered. Consequently, sharing his concerns with the oncologist resulted in a marked decrease in distress. Additionally, James immediately took up responsibility for the care of his partner. However, soon after, a metastasis was found, and Kelly had to undergo surgery again. Moreover, this time complications occurred and James had the feeling that he was losing all control over the situation. James described this period as "unreal" and at some point, he entered into a state of dissociation.

> "I sometimes feel like my emotions are going away. That may sound strange, but normally if I watch something emotional on TV, I get tears in my eyes. While now. . . People sometimes say that I pretend that nothing is going on here because I don't expose my emotions. It's not that I don't have emotions anymore, but I suppose that I shut them down a bit. Why? Well, I'm not the patient. She's the one who has the bad thing. I try to turn off my emotions [not to burden Kelly even more]."

To regain control over the situation and over his life, James took some days leave from work and went out for a walk as often as he could. When Kelly had an appointment with the oncologist and James was not allowed to join her, he put her in a wheelchair to get himself access to the hospital, determined to stay by his partner's side in case she received bad news. Moreover, whenever James felt the need to vent, he could call on his two best friends.

> "I have two real friends. They are my friends since my youth. Whenever I need to vent, I call them. Kelly knows them too, they're her friends too, and she knows that I talk to them about her. I can complain to them, I can whine about this and that, about anything. I can always rely on them."

Since the follow-up scans did not reveal tumor growth nor new metastases, Kelly regained her positivity and went back to work despite suffering from continuous neural pain. James was happy to see that his partner felt much better, but he kept on struggling with his own insecurity and doubts about the cancer and the unexplained flare ups of pain. Although he often felt helpless, he tried to be strong to avoid any conflicts and to hide his emotions not to upset his partner and children.

Since there was no cancer progression, the anxiety and insecurity disappeared steadily over the next months. Stimulated by his partner's positivity, James regained his faith in a future together. Moreover, James, who described himself as an ambitious workaholic, actively searched for ways to lower the work-related stress. Throughout the cancer process, James demonstrated personal growth. Indeed, he had become more tolerant, more patient, and more empathic. His priorities shifted from being successful and making money to being socially engaged and spending time with his family.

> "I can mostly enjoy small things now, such as going for a walk together, a bike ride, having a drink, even running errands together. That's what I learned to appreciate much more than I ever thought possible. I didn't use to do these things, and now, I like it all. I've learned that pleasant things don't have to cost much or don't have to be crazy. We were both extremely ambitious. That has been tempered. All that so-called show [appearance], those big cars, that's not important. That's not what life is about."

*Summary of other cases.* The participants in the gradually adapting resilience trajectory–James, Lilian, Michael, Norah, and Claire–were reliant on some individual characteristics that require interaction with others to foster resilience, such as 'being the information processor'–meaning that they took control over incoming and outgoing information about the cancer–and 'adaptive dependency', meaning that they were willing to ask for and accept help from others. These participants all showed a strong need to be seen and recognized as a partner of a dying person. Consequently, the availability of a support network that adhered to some basic rules, such as respecting autonomy and constantly adapting to the patient's partner's needs, was important for them. These internal and external resilience-promoting resources supported the partners in taking up responsibility, maintaining normality and routines in their daily life, and in mastering the situation.

**Ideal-type 3: Slowly adapting resilience.** This ideal type was characterized by severe distress for several weeks, followed by long periods of moderate distress typically lasting several months, ending in a stable status of mild distress, without returning to normal functioning. Nevertheless, there was also evidence for personal growth in these individuals. Moreover, clear signs of recovery and resilience after their partner's passing were reflected in the participants' stories.

*Optimal case*. Jasmine, a woman in her thirties and mother of two young children, recently lost her father and her best friend to cancer. When her husband William was diagnosed with brain cancer stage 4 and received a prognosis of two years at most, her world seemed to collapse. She developed severe physical and psychological symptoms and was temporarily unable to work. However, she immediately took over the household chores and the full care of the children. This motivated her to go on.

"Life at this point is extremely difficult. However, with children, one must go on. And I'm extremely grateful to our kids for that."

While unsolicited advice or stories of hope increased the level of distress even more, empathy and recognition as a partner of a dying person was much appreciated.

"He has those electrodes leading to a backpack and he's bald. But that's okay. It helps me to talk about it [the cancer]. Everybody can see that something is severely wrong, so yeah, it doesn't have to be hushed up or anything."

Jasmine could not fully enjoy beautiful moments with her family since these moments came along with the thought of having to miss these precious moments in the near future. However, it comforted her to hear her husband say that he accepted the situation, said he had had a good life so far, and felt happy despite the cancer. While Jasmine had doubts about her relationship before, she realized how many beautiful moments they had already experienced together and how much she would miss William.

Against all expectations, a revolutionary, experimental therapy succeeded in stabilizing the cancer for a long time. William saw the chance to finish his bucket list, and Jasmine picked up her normal life again. She went back to work, invited friends to her house, and organized family outings. The anxiety and negative thoughts were still present, yet less prevalent. As time went by, the need returned for Jasmine to have some time for herself. She even allowed herself to express frustration about her relationship with her husband. Although it was difficult not to be overwhelmed by grief and negative thoughts every now and then, life tended to become 'normal' again. For the first time in two years, Jasmine stopped saying goodbye constantly.

"We are also doing things from his bucket list. It's only natural for me to go along with this. We went to New York without the kids. He also went back to work one day per week and he's very pleased with that. So now we have something else [than cancer] to talk about. We all have our own little world again and I must admit, it feels normal. But we don't know how long it will last. Although I know that it makes no sense to think like this, I fear the day that this will come to a halt again. I can enjoy moments with my family again, mainly because there is some hope. I'm no longer saying goodbye in everything I do the way I did before. I always thought that what we were doing together would be the last time. But now, maybe it's not the last time, and that's worth so much."

However, when the couple was confronted with what could be bad news (a new spot on a scan), they both became very anxious again. However, this time, the physical symptoms were less prominent, and a sparkle of hope was maintained. Nevertheless, Jasmine sought professional psychological help for herself.

Two years after the diagnosis, the medical experimental therapy was still working, and William felt well. Jasmine kept searching for an equilibrium between allowing herself to be more

hopeful and being realistic. Furthermore, the uncertainty about the prognosis was difficult to cope with.

> "The feeling is double. It helps that there is no deadline anymore, but too much hope, is just... that's naive. That's how it feels. Even if you think that you're getting more time, you shouldn't start hovering."

Throughout the process, Jasmine learned a lot about herself. It became easier to ask for and to accept help, she became more empathic, and she learned to appreciate being a mother. Nevertheless, feelings of anxiety and distress were never far away.

*Summary of other cases*. The absence of positivity at the start of the process in all the participants from the slowly adapting resilience trajectory–Jasmine, Taylor, Audrey, and Meredith–is striking. Moreover, every positive thought was immediately followed by a negative one. Furthermore, as in gradually adapting resilience, being seen and recognized as the partner of a dying person was essential for partners in this group. As a result, they became more distressed whenever their partner looked better because people then tended to wrongly assume that their partner was healthy. However, a more positive attitude, flexibility, and an inner strength emerged throughout the process in all these partners, albeit at a slower pace than in the participants of the gradually adapting resilience trajectory. These individual resilience-promoting resources allowed for taking up responsibility, maintaining normality in daily activities, and even mastering the situation. Despite these adaptive coping strategies, the level of distress remained moderate to mild throughout the process. Although some personal growth occurred throughout the process, the interviews of these participants did not evidence a return to pre-diagnosis levels of wellbeing.

**Ideal-type 4: Continuing distress.** The advanced cancer diagnosis immediately evoked severe distress that persisted or gradually turned into a long-lasting moderate distress. Throughout the trajectory, there were no signs of personal growth and the participant showed no signs of post-traumatic growth. On the contrary, the trajectory was characterized by unprocessed anger.

*Optimal case*. Louisa and her husband Pete both lost their parents at a young age. They had many common interests and hobbies and were described by others as a 'symbiotic' couple. Indeed, they rarely did anything alone. Yet, they also had difficulties expressing their emotions to one another and to others, and Louisa remarked that it seemed that they experienced their feelings only as 'subdued'. When Pete was diagnosed with advanced cancer, their lives were turned upside down. Pete immediately talked about euthanasia. This idea terribly upset Louisa, causing much anxiety as well as anger. When Pete decided to undergo chemotherapy, Louisa regained some hope. Nevertheless, she called that period "the worst that one could ever experience". Louisa decided not to share her fears and concerns with Pete nor with others. She desperately tried to continue ordinary life and she did not talk about the cancer with anyone, not even with her husband.

> "We have a lot of upcoming events. There's a grandchild on the way and a wedding is planned. He cannot die now. He's still so much needed here. And yes, you hear it that much, people who have cancer for a long time, who live for four, five, six years. One should always have hope, right?"

When new metastases occurred and Pete was admitted to the hospital, Louisa felt angry and moderately distressed. The distress increased even further when Pete developed anorexia. She tried to manage the cancer herself and asked the doctors to administer artificial nutrition.

Since her request was immediately denied, she felt anxious and suffered from nightmares and migraines.

> "A lot of things go well, but there are other things that don't go well. For instance, eating is extremely important, but he can't swallow well. Food can hardly pass through his throat. So, I started with some fresh cheese and said: 'C'mon, eat something'. But he was admitted to the hospital, and he was so faint I had to hold him. And my daughter also said: 'Mom, you should insist on tube feeding'. But they didn't . . . so, I called my daughter to tell her they didn't want to do what I asked. Hence, we keep on giving him two spoonful of fresh cheese and I put all kind of food in front of him, but he can't even drink water."

Since Pete had always been the only one Louisa relied on, she did not dare to ask help from her adult children once Pete was no longer there to support her.

> "I had a husband who always had the best ideas. Every time I didn't feel well, he said: 'C'mon, let's do this or that'. But now, I'm standing all alone in front of everything, right? I have two children, but you know how things go, they have their own families. It feels more like, you'll have to do it all on your own. I can't just call my children and ask them what I should do. Not for small things and not for big matters. Maybe they could give some advice, but in the end, I can no longer make consensual decisions [with her husband], right?"

Six months after Pete's death, Louisa did not show any signs of recovery or resilience. On the contrary, she felt desperate and distraught. Despite the daily presence of her supportive family and her beloved grandchildren, she felt lonely and lost. She cried every day and described the past few months as "a succession of misery".

*Summary*. In this case, dependent attachment to the partner was apparent, with high levels of anxiety, anger, and despair. The partner tended to rigidly stick to daily routines and attempted to take control over the cancer by demanding pointless treatment such as artificial nutrition. Although a support network was available, Louisa was not able to use this support or truly benefit from it. Moreover, the frenetic efforts to maintain daily life kept her from adapting to the circumstances.

**Ideal-type 5: Delayed distress.** The diagnosis of advanced cancer elicited mild or moderate distress in the immediate aftermath of the diagnosis and was followed by a stable period of no distress. However, high levels of distress arose seemingly 'suddenly' (without a specific cause or new event) and continued over time.

*Optimal case*. When Dorothy's partner Phil was unexpectedly diagnosed with cancer, she felt highly distressed. However, when he relapsed several years later and was diagnosed with cancer in an advanced stage, it hardly seemed to distress her. She found herself well prepared, placed the responsibility for healing on the oncologist, had faith in new therapies, and felt ensured that everything would turn out well again.

> "The oncologist had already notified us that it could go fast. In contrast to the first cancer diagnosis, we were prepared this time. I already know what difficulties I will encounter, and what he will have to endure during the treatment. Hence, that makes it all much easier. The first time was different because I wasn't prepared at all, but now I am, and I can accept it all. And we have a wonderful doctor too. She discusses every step in the treatment with us and she even mentions the word 'dying'."

Her positivity and inner strength made her take up the responsibility for both her and her partner's wellbeing by resolutely focusing on how lucky they were and how grateful they should be for living in a country known for its excellent health system.

"What we both did and still do is telling each other how lucky we are. We sometimes ask ourselves how people dare to complain. We are safe here, we have a roof over our heads, we have enough food, we have clothes, we can warm ourselves, and we are treated. Moreover, his treatment costs us nearly nothing. Everything is reimbursed by the government. Some people complain about COVID. They have no right to do so. We all have so many benefits here. That's something we repeat regularly."

When thoughts about the future arose, she focused on what she could continue despite what happened, how life would stay largely unchanged, and convinced herself that she might die even before her partner and would not have to endure widowhood at all. Only when her partner was being hospitalized for a bone marrow transplant did she have the feeling of losing control over the situation and felt briefly distressed. She responded immediately to these negative feelings by visiting friends and family. As such, she avoided being alone and thus was distracted from the situation, which helped her in regaining mental strength.

"The hardest moment was when I had to drop him off at the front door [of the hospital]. I wasn't allowed to enter, and that was terrible. He waved and was gone. I stood there. I went immediately to my son's, where I crashed, crying. I stayed there for two days. From there, I went to a friend's for another two days, and then to another friend's, and to my daughter's until he came back home. While he was in hospital, my daughter-in-law did the laundry, as everything had to be clean and sterile. Thinking back, I think that support was crucial. Normally, I like having it all under control myself, but when he was hospitalized, I couldn't do anything, I was even not allowed to visit him."

As soon as her partner was discharged from hospital, they left for a long trip abroad and only returned once a month for the partner to receive immunotherapy. Living with cancer became a new way of life in which the cancer was present only in the background. However, more than two years after the patient was diagnosed with advanced cancer, and after six months of complete remission, she realized that, if her partner relapsed a second time, there would be no treatment options available anymore. The uncertainty about the prognosis–nobody could tell her what her partner's prognosis was, as he was already living on 'borrowed time'–frightened her and she felt that she was losing control over the situation. When she looked at the future, she could only see an uncertain, blurred one without her partner. She tried to talk positively to herself but was overwhelmed by existential questions without answers. Feelings of distress were increasing quickly, and an anxiety disorder was diagnosed.

*"Over the last few weeks or months, for a while anyway, I have those anxiety spells. Anxiety is a big word, but some overwhelming thoughts. What if I am left alone? Without him? So, I say to myself: 'Yes, what would you do? You can do this and that'. But that's something that bothers me and that keeps me awake at night. I don't want this, but yeah, it's a kind of freewheeling of my brain and then I think: 'Yes, if you would be alone . . .' That's perhaps the most problematic thing about the cancer."*

*Summary.* Dorothy showed a positive attitude and inner strength that was expressed as seemingly being able to master the situation without external help. Yet, as in the continuing

distress trajectory, there was a marked absence of flexibility, and she rigidly tried to maintain normality in daily life. However, as distress started to increase (e.g., when her partner was hospitalized), the availability of a support network became more important and was used to distract from the cancer. It was only after the partner seemed to be cured from the cancer that Dorothy increasingly lost control and her defense system against feelings of anxiety and despair began to fail.

**Ideal-type 6: Frozen disconnection.** This trajectory was characterized by a dissociation from the experience, implying an inability to adapt or change in response to the patient's diagnosis of advanced cancer. The (mild) distress that occurred because of the advanced cancer diagnosis was responded to with rigidity instead of resilience.

*Optimal case*. When Douglas' partner Josie was diagnosed with advanced cancer, he was shocked and puzzled. However, the next day he decided not to allow the cancer to affect his own life. He rigidly attempted to maintain everyday life meanwhile almost ignoring the cancer.

> [A few days following the diagnosis of metastatic cancer and some weeks before the start of Josie's chemotherapy]. "The first day [of the holiday] we were sitting there with a glass of wine, and I thought: 'We're doing well'."

When Josie could no longer participate in certain activities and was forced to adapt her life to the cancer, Douglas chose to continue the activities alone.

> "After the surgery, we tried, to the best of our ability, to maintain a normal life. Yes, by doing all kinds of things. I also went on a weekend trip myself [with friends], without her. But I did this before the cancer too. That wasn't anything new. So, we just tried to go on as usual as well as we could."

He was grateful that Josie never complained and was surrounded by her own family and friends, a group that he did not consider his family and friends. Difficulties at work were much harder to deal with and caused more distress than his partner's cancer diagnosis.

> "My work has brought me into a burnout before, more than once actually. And some months ago, I came home, and all had been too much again [at work]. I didn't know what to do anymore. It was the last day before my holidays, but despite this, it was all too much. After a three weeks' vacation, I went back to work but only for a few days. Then I had to take sick leave from work for another month."

*Summary*. By analogy with types 4 and 5, the participant's trajectory was characterized by an absence of flexibility, resulting in rigidly attempting to maintain every aspect of daily life unchanged. The participant disconnected from the cancer by dividing their life as a couple into the lives of two individuals. As such, the cancer became the problem of his partner and her friends and family while he himself, his friends, and his family could observe the events from a distance.

## Wellbeing

Fifteen out of the 17 participants filled in the bi-monthly Dutch Mental Health Continuum-Short Form (MHC-SF) on at least two occasions [15].

There seemed to be no apparent association between overall wellbeing scores, specific subscales of the wellbeing questionnaire (emotional, social, and psychological wellbeing), and the

identified ideal-type trajectories. The lowest scores in terms of overall wellbeing could be found in the participant from the continuing distress trajectory while the highest scores in overall wellbeing were reported by the participant of the delayed distress trajectory. Furthermore, the lowest scores in social wellbeing were seen in the participant from the frozen disconnection trajectory.

A graphic representation of the scores from the MHC-SF by participant, along with the followed trajectory based on the estimated levels of distress, is provided as S4 File.

## Discussion and conclusion

Applying an ideal-type analysis on the longitudinally obtained data from 54 interviews conducted over three years in 17 participants revealed six trajectories of adjustment to an intimate partner's diagnosis of advanced cancer, namely: rapidly adapting resilience, gradually adapting resilience, slowly adapting resilience, continuing distress, delayed distress, and frozen disconnection. The first three mentioned trajectories included varying levels of resilience, as they each reflected different ways of adapting well in the face of adversity, trauma, tragedy, or threats [20].These trajectories revealed a return to levels of functioning before the PTE, sustainability (to be able to persevere), and growth (gains and advancements through new learning and attainment of inner strength) [21, 22]. The remaining three trajectories, however, reflect a less optimal adjustment, although each of these also reflects clear attempts at adaptation, but they seemed to be less successful in terms of overall functioning.

The four most common trajectories of adjustment following a variety of PTEs, which have been identified by broad-ranging studies using latent growth modeling (resilience, recovery, delayed reactions, and chronic stress) [6, 7], also emerged in this study. By analogy with the so-called minimal impact resilience trajectory–meaning that adults respond to a PTE with minimal disruptions in overall functioning [6]–participants following the rapidly adapting resilience trajectory responded to the PTE, after a brief period of distress, by actively and inventively finding ways to cope with their partners' diagnosis of advanced cancer with few if any disruptions in everyday functioning. Furthermore, similar to what is considered a recovery trajectory [6], our participants from the gradually adapting resilience trajectory responded to the partners' advanced cancer diagnosis by acute moderate to severe distress, significantly influencing their physical or psychological functioning. Over time, their stories demonstrated a level of personal growth that helped them return gradually to healthy functioning without long-lasting symptoms of distress. Consistent with previous studies on resilience [10], resilience trajectories were also the most common in our group of participants.

Two of the less optimal trajectories, namely continuing distress and delayed distress, are likewise described in a minority (5 to 30%) of individuals following different PTEs, such as terrorist attacks, bereavement, hospitalization with COVID-19, etc. [6, 23]. From this study it cannot be inferred whether the participant who followed the continuing distress trajectory would have developed resilience-supporting individual characteristics and would have followed a slowly adapting resilience trajectory if her partner would have had a slower disease progression, allowing her more time to adapt to the situation. Additionally, despite relying on resilience-promoting resources, those from the slowly adapting trajectory needed more than average time to regain a status of mild or moderate distress. Moreover, either they did not seem to reach a status of no distress before death of the partner, or they could still be considered mildly or moderately distressed at the end of the study. To the best of our knowledge, this trajectory has not yet before been described in resilience research following a PTE. Nevertheless, longer follow-up periods are needed to investigate whether or not this trajectory transitions into a gradually adapting resilience trajectory when the patient's condition becomes

more chronic. At any rate, a slowly adapting resilience trajectory does not hamper a resilient outcome as was demonstrated by our participants. Additionally, dissociation as well as the freeze response are well known phenomena in stress research and studies with a psychopathology approach (e.g., compassion fatigue, PTSD) [24, 25]. Meanwhile, the frozen disconnection trajectory–meaning that one disconnects from the cancer without signs of adaptation–has not been described in research on adaptation to a PTE before. Indeed, disconnection and being unable to react are usually seen as maladaptive coping strategies [26]. Nevertheless, a freeze response can prepare someone for action and can reduce the impact of the PTE [27], while it can be assumed that dissociation could be helpful to tolerate stress or intense emotions.

Our findings underpin and extend the conclusions of a former meta-synthesis on resilience in advanced cancer caregivers [4]. Certainly, the resilience trajectories are promoted by personal characteristics (flexibility, positivity, inner strength, the ability to control incoming and outgoing information, and to ask for and accept help) and by the availability of a support network [4, 11]. In the current study some individual resilience-promoting characteristics were more often associated with specific trajectories. In fact, the characteristics that do not involve interaction with others (flexibility, positivity, and inner strength) were more common among participants with a rapidly adapting resilience trajectory, while participants with a gradually adapting resilience trajectory were more reliant on the interactive resilience characteristics, such as the ability to control incoming and outgoing information and to ask for and accept help. Nevertheless, all resilience trajectories were characterized by the participants demonstrating a personal growth process in which the benefits found throughout caregiving often fueled the existing individual characteristics, or even allowed new characteristics to emerge, giving the resilience trajectories a dynamic aspect. Consequently, this allows us to endorse the American Psychological Association (APA)'s assertion that resources and skills associated with positive adaptation to adversity can be cultivated and practiced [20]. Furthermore, all participants relied on a support network, albeit in different stages of the resilience trajectories. We can confirm the importance in the development of the patient's partner's resilience trajectory of the availability of a support network (family, friends, and healthcare professionals) adapting flexibly to the changing circumstances as the patient's prognosis worsens [12]. Indeed, being disregarded in the role as partner of a patient with an unfavorable prognosis proved to be one of the most severe causes of prolonged distress and even hampered the expression of individual resilience-promoting resources.

Beyond that, we can note that all our participants were confronted with more than one PTE simultaneously. In fact, not only was their partner diagnosed with advanced cancer during the COVID-19 pandemic, most participants also faced the recent loss of a loved one or faced serious or life-threatening conditions in family members or friends. Despite this, most followed a resilience trajectory. This confirms the findings from a previous study on resilience regarding the occurrence of two or more PTEs concurrently [13].

## Implications for practice and future research

Long-term trajectories of mental adaptation following a PTE, such as the partner being diagnosed with advanced cancer, can be diverse. Moreover, the diagnosis of advanced cancer often means the onset of a series of stressful events, such as the occurrence of new metastases, hospital admissions, or uncertainty about prognosis, all of which require adjustments at each point. If further replicated, the six identified ideal-type trajectories of resilience could support HCPs in distinguishing those advanced cancer caregivers in whom a resilient outcome can be expected, and those who might struggle more and may need more intensive follow-up. However, it is necessary to regularly reassess the caregiver's trajectory since it cannot be ruled out

that new events may cause one trajectory to transition into another. Additionally, although the description of the ideal types could suggest a categorical distinction, the resilience trajectories should rather be considered as different parts of a continuum between rapid and slow adaptation to the patient's advanced cancer diagnosis. Consequently, in clinical practice, caregivers could often align with more than one ideal-type trajectory.

Furthermore, resilience-promoting resources, namely personal characteristics and the availability of a support network, should be considered since they could promote a resilience trajectory. The implications for practice in this regard suggest that our efforts should be directed towards stimulating available resilience-promoting resources through psychological counseling [28]. Nevertheless, the presence of sufficient resilience-promoting resources still cannot guarantee a resilience outcome. As such, a rapidly adapting resilience trajectory always entails the risk of delayed distress. More longitudinal studies with a more extensive follow-up period are needed to gain full insight into when, why, how, and by whom one trajectory transitions into another and what the outcome implications are.

Moreover, future research could delve deeper into baseline adjustment factors that may impact resilience trajectories in cancer caregiving, including personality traits, attachment styles, past significant life events, cultural norms, spiritual beliefs, and other traditions, all of which play crucial roles in individuals' emotion regulation, interpersonal dynamics, and coping mechanisms [29]. Finally, it is intriguing that there was no clear relationship between the self-reported values of wellbeing (in the MHC-SF) and the different resilience trajectories. This could be explained by individuals tending to enhance well-being measures based on social desirability, resulting in response artifacts [30]. However, this could also indicate that mental health (measured in the MHC-SF) and mental distress (inferred from the interview data) are two separate concepts and not two endpoints of one continuum [31]. Further longitudinal mixed-method research tracing trajectories of wellbeing could shed light on this.

## Strengths and limitations

Conducting repeated interviews allows for collecting rich and detailed data and capturing change and development of the studied phenomenon [32, 33]. Hence, a longitudinal qualitative study aligns best with studying trajectories [33]. Moreover, adaptations of the context, though often subtle, can be mapped by repeatedly conducting interviews [33]. Also, interviewing participants every six months ensures data collection close to the time of occurrence and consequently avoids recall bias [34].

Nevertheless, this study was also prone to certain limitations. An ideal-type analysis implies condensing large amounts of interview data into narratives and abstracting data to construct the ideal-type trajectories [14]. As a result, details may be omitted in the process. In addition, participants who deliberately decide to take part in a longitudinal study on resilience may have specific characteristics and may not be representative of the population. Furthermore, some participants only agreed to a subsequent interview when the patient's condition was stable, which could lead to a too positive narrative. Likewise, details from one's history could only be obtained in a retrospective manner, likely leading to positive recall bias [35]. Nevertheless, three out of 17 participants exposed a less preferable trajectory, while the others could be considered resilient. This is consistent with findings from quantitative studies on resilience in other domains [5, 6, 9]. However, the three less preferable trajectories were represented by only one case each, implying the risks of limited nuance and incomplete trajectory description. Furthermore, although a history of depression or current treatment with antidepressants was an exclusion criterion, an underlying, undiagnosed depression or anxiety disorder in the

participants could not be ruled out. As a result, the influence of any resulting negative psychological defenses on the course of the study could not be excluded.

Although all the participants were adults under 65 years of age at the start of the study, they were not all in the same life stage. Therefore, it is unclear whether the type of trajectory is associated with stage of life, having a family with young children, or engaging in an active professional life. Moreover, despite actively searching for candidates of non-European origin, all our participants were Flemish. Additionally, despite the use of an extensive interview guide, information about the participants' backgrounds, which is crucial to fully understand one's trajectory, is sometimes lacking as people do not always associate specific events, habits, engagements, or cultural aspects with coping with advanced cancer. Moreover, by interviewing the participants twice a year, the researcher inevitably develops relationships with the participants over time which can introduce bias into data collection. To avoid researcher bias, all interviews were analyzed by two researchers from different backgrounds independently and discussed on several occasions within the research team [36]. Finally, data collection for this study took place mainly during the COVID-19 pandemic. Therefore, the influence of the pandemic on the results should not be underestimated. Indeed, the pandemic and the measures taken to prevent the spread of the virus could be considered a (second) PTE and thus as an unusual stressor, implying the risk of severe mental health effects and higher levels of distress [13].

## Conclusion

This study provides novel insights into the understudied domain of resilience in cancer caregiving. In response to the diagnosis of advanced cancer, six ideal-type trajectories of adaptation could be distinguished in the partner of a patient with advanced cancer. Three trajectory types (rapidly adapting resilience, gradually adapting resilience, slowly adapting resilience) are considered resilient processes and entail a high probability of a resilient outcome, such as a stable status of mental wellbeing or even personal growth. In contrast, three trajectory types (continuing distress, delayed distress, and frozen disconnection) are likely to provide a less preferable outcome.

Resilience-promoting resources as described in former studies may determine the direction of the patient's partner's trajectory since participants who could rely on a set of individual characteristics such as positivity, flexibility, and inner strength were more likely to follow a rapidly adapting resilience trajectory. Meanwhile, participants who had optimal control over incoming and outgoing information about the cancer and those who could rely on a well-established and adaptive support network were more likely to follow a gradually adapting resilience or a slowly adapting resilience trajectory. Nevertheless, the presence of resilience-promoting resources cannot always guarantee an optimal resilience trajectory nor a resilient outcome. Indeed, more research is needed to gain insight into the complex interactions between resilience-promoting resources and one's baseline adjustment in building a resilience trajectory.

## Supporting information

**S1 File. Semi-structured interview guides.** Interview guides for the first, follow-up, and final interviews. The interview guides were compiled with input from a partner of a patient recently deceased from cancer. The interview guides were slightly adapted after the initial interview. The version as published here was used for all subsequent interviews.
(PDF)

**S2 File. Degrees of distress.** Criteria for each level of distress as agreed in consensus by the authors.
(PDF)

**S3 File. Narratives and illustrating quotes organized by ideal-type trajectory.** Detailed narratives different from the optimal case, including contextual features and illustrating quotes for each trajectory of resilience.
(PDF)

**S4 File. Participants' scores on the MHC-SF.** Graphic representation by participant of the scores on the MHC-SF, along with the followed trajectory based on the estimated levels of distress.
(PDF)

## Acknowledgments

First and foremost, the authors wish to thank the participants for their willingness to repeatedly participate in the interviews during what was one of the most challenging periods of their lives. Furthermore, we want to thank the oncology teams of the University Hospital Leuven and University Hospital Ghent, the GPs of the Leuven north region, and the managers and board members of the peer groups for those with advanced cancer for their help with the recruitment of the participants. We also wish to thank Mrs Iris Vande Walle for her assistance with drawing up the study protocol and the interview guide and for her critical reflections on the descriptions of the trajectories. Additionally, we wish to express our sincere appreciation and thanks to Dr Curt Dunagan for his linguistic support in the writing of the manuscript, as well as for his proofreading. Finally, we want to thank the University Foundation of Belgium (Universitaire Stichting van België) for supporting this publication.

## Author Contributions

**Conceptualization:** Sophie Opsomer, Jan De Lepeleire, Patrick Luyten, Peter Pype, Emelien Lauwerier.

**Data curation:** Sophie Opsomer, Luca De Clercq, Sofie Joossens, Peter Pype, Emelien Lauwerier.

**Formal analysis:** Sophie Opsomer, Luca De Clercq, Jan De Lepeleire, Sofie Joossens, Patrick Luyten, Peter Pype, Emelien Lauwerier.

**Investigation:** Sophie Opsomer, Jan De Lepeleire, Peter Pype, Emelien Lauwerier.

**Methodology:** Sophie Opsomer, Jan De Lepeleire, Patrick Luyten, Peter Pype, Emelien Lauwerier.

**Project administration:** Sophie Opsomer, Patrick Luyten, Peter Pype, Emelien Lauwerier.

**Supervision:** Jan De Lepeleire, Patrick Luyten, Peter Pype, Emelien Lauwerier.

**Validation:** Jan De Lepeleire, Patrick Luyten, Peter Pype, Emelien Lauwerier.

**Visualization:** Sophie Opsomer.

**Writing – original draft:** Sophie Opsomer.

**Writing – review & editing:** Luca De Clercq, Jan De Lepeleire, Sofie Joossens, Patrick Luyten, Peter Pype, Emelien Lauwerier.

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
