## [Decision Letter · Decision Letter 0]

26 Mar 2024

PONE-D-24-01787Do all roads lead to Rome? An ideal-type study on trajectories of resilience in advanced cancer caregiving.PLOS ONE

Dear Dr. Opsomer,

Thank you for submitting your manuscript to PLOS ONE. After careful consideration, we feel that it has merit but does not fully meet PLOS ONE’s publication criteria as it currently stands. Therefore, we invite you to submit a revised version of the manuscript that addresses the points raised during the review process.

We look forward to receiving your revised manuscript.

Kind regards,

Maria Berghs, PhD

Academic Editor

PLOS ONE

Journal Requirements:

4. Please ensure that you refer to Figure 2 in your text as, if accepted, production will need this reference to link the reader to the figure.

**Additional Editor Comments:**

The reviewers point to a need for some minor revisions and are both complimentary. I think their feedback can be incorporated, they give some useful advice and this will ensure a stronger paper.

Reviewers' comments:

Reviewer's Responses to Questions

**Comments to the Author**

1. Is the manuscript technically sound, and do the data support the conclusions?

Reviewer #1: Yes

Reviewer #2: Yes

2. Has the statistical analysis been performed appropriately and rigorously? 

Reviewer #1: N/A

Reviewer #2: N/A

3. Have the authors made all data underlying the findings in their manuscript fully available?

Reviewer #1: Yes

Reviewer #2: Yes

4. Is the manuscript presented in an intelligible fashion and written in standard English?

Reviewer #1: Yes

Reviewer #2: Yes

5. Review Comments to the Author

Reviewer #1: This is an important study that was also well written and was clear. I had a few minor comments

1. Within the methodology the authors are not explicit about the methods used.

2. One of the inclusion criteria was if the participant was fluent in Dutch, but it is not clear if the interviews were conducted in Dutch and if so then there should be statement of translation of the data

3. More information is needed in the reflexivity section on how the researchers could have influenced the findings of the research

4. The quotes of the participants should have the quotation marks

5. Regarding consent of the participants in the follow up interviews , were participants reconsented on the subsequent interviews or was it at the beginning only. What happened if the participants hade reservations on the next interview.

Reviewer #2: Thank you for the opportunity to review this well-written paper. I found it interesting to read. There are a few minor areas where clarity is required.

Aims & rationale: The aims of the study and rationale for the study are clearly outlined – and the originality of the study and the novel contributions it makes to the existing literature are made clear. Lines 45-47 – it is stated that it is assumed that the approximate 60% resilience/recovery rates found in the research on PTEs apply to partners of those with an advanced stage diagnosis…”since in clinical practice, most of these partners have a resilient outcome”. Is this the anecdotal/subjective experience of the authors? If so, this should be stated for clarity’s sake (otherwise the sentence requires a reference). This is then referred to later in the discussion where the findings of this study do indeed support this assumption.

Methods: The methodological approach used in the study is clearly explained – and sufficient measures are taken to ensure consistency in analysis between researchers, which enhances the reliability and validity of the study. Clarity is required with regards to participant completion of the Dutch Mental Health Continuum-Short Form (MHC-SF): lines 104-106 do not appear to match what is stated in lines 552-553. This needs to be consistent.

Findings and Discussion: The findings are presented logically and suitably evidenced. I found the arguments made convincing. The discussion locates the findings clearly in the existing literature.

This does not need to be addressed in the paper (as it is addressed in the implications section – where further research on ‘personal characteristics’ is recommended), however, for interest’s sake, what I found fascinating is that results line up with the understandings offered by psychoanalytic and attachment theories too. The ‘participant characteristics’ referred to as positivity, flexibility and inner strength, could be understood from related theoretical orientations to represent participants’ mental health, personality structures, ego strength, emotion regulation capacities and interpersonal relationship skills, which are influenced by their personal histories of previous loss/es. Current findings suggest that across cultures approximately 60% of the population is securely attached, with the remaining 40% exhibiting insecure (preoccupied/dismissive) attachment or disorganised/unresolved attachment. These attachment states of mind reflect styles of psychological defence and strategies for managing emotional threats, in addition to capacities to use interpersonal support. Those with insecure and disorganised attachment states of mind tend to exhibit over- or under-regulation of emotion and the use of more immature defences such as denial, which was found in the continuing distress and delayed distress ideal types. They had denial in common – that when breached by death or the reality of likely loss – then appeared to cause emotional distress. The frozen disconnection ideal type was suggestive of disorganised attachment where dissociation as a defensive strategy is common. This type seemed to have the most rigid denial defence – to the point that even when distress emerged, it was still dissociated/denied from the cancer of their partner and misattributed to other causes e.g. work burnout. Incidentally, the fact that the self-report questionnaire findings did not appear to correlate with reviewer observations of distress may reflect these very distortions caused by intrapsychic defensive processes. Healthy help-seeking is also supported by these theories, as the distortions in thinking found in insecure/disorganised attachment states of mind influence capacity for mature relating to others.

One clarification in this section related to line 444 where it is stated: “strong attachment to the partner was apparent” – this could be misconstrued as a strong attachment, psychologically speaking, is good/healthy – this attachment is perhaps better characterised as dependent?

Limitations of the study: Although the effects of COVID-19 as an additional stressor are mentioned, the fact that references to the pandemic are scattered throughout the evidencing quotes, suggests that its influence may have coloured the findings more significantly than acknowledged. This was a largely unusual worldwide stressor that had unique mental health impacts, and the unique uncertainty it posed to those with existing healthcare conditions, the uncertainties surrounding access to healthcare, and the impact that restrictions on visitors/accompanying support to healthcare visits/hospitalisations had on the mental health of the ill and their families is noted in other literature. It might not be sufficient to consider it as merely an additional PTE. It should at least be stated that the fact that data collection occurred during the pandemic may have influenced the findings of the study. Also, I wondered about methodological limitations related to the existing mental health of participants – although a previous depression diagnosis excluded potential participants, I wondered whether the continuing distress ideal type had unresolved loss and poor emotion processing capacities to begin with (perhaps existing low level, undiagnosed mental health concerns). The example used indicates overuse of dismissive psychological defences, such as denial, suppression and repression. Thus, the exclusion criterion of an existing diagnosis may not have controlled sufficiently for existing mental health concerns.

Minor typo:

Line 599 – ‘chronical.’ Should be ‘chronic’

6. PLOS authors have the option to publish the peer review history of their article (what does this mean?). If published, this will include your full peer review and any attached files.

Reviewer #1: No

Reviewer #2: No

---

## [Author Response · Author response to Decision Letter 0]

22 Apr 2024

We hereby confirm that we have reviewed the style requirements and adapted the layout of the manuscript accordingly. 

The manuscript was proofread again. To improve readability and comprehension, a few sentences have been slightly rephrased, however without changing the meaning. 

In fact, this manuscript is subject to some ethical restrictions on sharing a (de-identified) dataset. The interview data contain highly sensitive information and include indirect identifiers. As mandated by the ethics committee of the University Hospital Leuven, the interviews cannot be published for reasons of privacy and confidentiality.

We updated our Data Availability statement as follows:

This paper employs a ‘minimal data set’ (consisting of narratives and illustrating quotes) in order to highlight the conclusions drawn. However, the interviews contain potentially highly sensitive information and involve indirect identifiers. For privacy and confidentiality reasons and mandated by the Ethical Commission of University Hospitals Leuven, the interviews and/or the interviewer’s field notes cannot be published or distributed. The complete datasets used and analyzed during the current study are deposited in the repository of the Academic Center of General Practice (ACHG), KU Leuven and are available from the corresponding author or from the head of the department, Prof. Dr. Birgitte Schoenmakers, Kapucijnenvoer 7 / 7001, 3000 Leuven (birgitte.schoenmakers@kuleuven.be) on reasonable request, and after the inquirer can provide written permission by the Ethics Committee Research UZ / KU Leuven, Herestraat 49, 3000 Leuven (ec@uzleuven.be). The interview transcripts will be delivered in the original language (Dutch) and will be anonymized and de-identified. 

The ethics statement in the methods section of the manuscript was extended with the information originally mentioned in the declaration section. We removed the ethics statement from the declaration section. The Ethics section, as a subheading in the Methods section, reads as follows: 

The interviews were conducted according to the COVID-19 measures in force at that time. 

This study adhered to ethical standards established by institutional ethics committees and the principles outlined in the 1964 Helsinki Declaration for research involving human participants. All participants took part voluntarily in the study. Ethical approval was provided by the Ethics Committee Research UZ / KU Leuven on October 4, 2019, study number S63166; by the Ethics Committee of Ghent University Hospital on October 17, 2019, study number BC-06066; and by the Ethics Committee of Imelda Hospital Bonheiden on June 9, 2020. An amendment to extend recruitment to general practices and peer group websites of oncology patients was approved by EC UZ/KU Leuven on July 7, 2020 and by EC Ghent University Hospital on July 14, 2020.

Prior to commencing the study, written informed consent was obtained from all individual participants and their partners. The consent included permission for publication of anonymized and translated quotes from the interviews. Although the written consent covered all interviews, oral consent was also obtained before each interview. Participants were informed, both in writing and orally, that they could withdraw from the study at any time without providing a reason. 

Furthermore, participant data has been anonymized in order to preserve the scholarly meaning while protecting the privacy of the individuals involved. 

(Line 209-228)

4. Please ensure that you refer to Figure 2 in your text as, if accepted, production will need this reference to link the reader to the figure.

Fig 2 is referenced in the tekst as follows: (Line 244)

A graphical representation is provided as Figure 2. (Line 244)

Fig. 2. Trajectories of resilience. A graphical presentation of six prototypical trajectories of adjustment to an intimate partner’s diagnosis of advanced cancer: (A) Resilience trajectories; (B) Non-resilience trajectories. (Line 247)

We are not aware of quoting a retracted article. 

However, to be sure, we looked up all articles from the full reference list. We can hereby confirm that our reference list contains no retracted articles.

The reference list has also been checked for completeness. Since some statements were not properly referenced, we added some additional references (ref. 32-34).

As a result of an expansion of the text requested by the reviewer, 3 references were added (ref. 17-19). 

We can confirm that our reference list is now complete, correct, and applies to the Vancouver style. Where applicable, the doi of the article was added. 

Additional Editor Comments:

The reviewers point to a need for some minor revisions and are both complimentary. I think their feedback can be incorporated, they give some useful advice and this will ensure a stronger paper.

We fully agree with the editor that the reviewers’ comments were very useful and constructive. They made us think about some paragraphs and helped us to approve the manuscript. 

Reviewers' comments:

Reviewer's Responses to Questions

Comments to the Author

5. Review Comments to the Author

Reviewer #1: 

This is an important study that was also well written and was clear. I had a few minor comments

We would like to thank you, reviewer 1, for reviewing our manuscript and for your useful and constructive comments. 

Within the methodology the authors are not explicit about the methods used.

To clarify the methods used, we have added the following paragraph to the Methods section:

A longitudinal qualitative study was conducted with seventeen partners of patients diagnosed with advanced cancer. To identify trajectories of resilience or lack thereof, ideal type analysis - a method for constructing typologies from qualitative data - was used to analyze data from fifty-four interviews conducted over a three-year period. (Line 81-84)

2. One of the inclusion criteria was if the participant was fluent in Dutch, but it is not clear if the interviews were conducted in Dutch and if so then there should be statement of translation of the data

Indeed, all interviews were conducted in Dutch and were not translated to English except for the quotes we used to illustrate the findings.

We added: All interviews were conducted in Dutch to the data collection section (line 114). 

We added: ... and translating the illustrative quotes from Dutch to English to the data analysis section (line 166).

3. More information is needed in the reflexivity section on how the researchers could have influenced the findings of the research

We added the following paragraph to the trustworthiness section.

The quality of qualitative studies is judged by their trustworthiness which includes several criteria such as credibility, transferability, reliability, and reflexivity [1, 2]. In a longitudinal qualitative study trustworthiness can be challenged by researcher bias as the interviewer and the participants inevitably get to know each other through repeated interviews. This relationship may affect the depth and quality of the data obtained [3]. In addition, credibility may be threatened because researchers may have preconceived notions that affect how they interpret data and draw conclusions [3]. However, to ensure the trustworthiness of the study by increasing reflexivity, the interviewer’s field notes were considered. (…) In addition, to optimize the relevance, credibility, and completeness of the data, the study protocol and interview guide were developed in consultation with someone who had experienced the loss of a partner to cancer. To further ensure trustworthiness, credibility, and transparency, interviews from the first three cases were analyzed by three researchers independently. (Line 170-177 and 180-191)

4. The quotes of the participants should have the quotation marks

We added quotation marks to the quotes and removed the italics. 

5. Regarding consent of the participants in the follow up interviews , were participants reconsented on the subsequent interviews or was it at the beginning only. What happened if the participants hade reservations on the next interview.

The written ICF included information and consent for the first and all subsequent interviews. However, participants were invited to subsequent interviews and asked if they agreed to be interviewed. Therefore, verbal consent was given before each interview. The ICF also included the information that one could decide not to participate at any time, even during an interview. One participant decided not to have a second interview because she didn't want to talk about cancer anymore. Since she only had one interview, we excluded her from the study and did not analyze the interview. Another participant decided to drop out just before the fourth interview (after almost two years in the study). He preferred not to give a reason for dropping out. His three interviews were analyzed and are part of the study. 

We have added the following paragraph to the ethics section:

Prior to commencing the study, written informed consent was obtained from all individual participants and their partners. The consent included permission for publication of anonymized and translated quotes from the interviews. Although the written consent covered all interviews, oral consent was also obtained before each interview. Participants were informed, both in writing and orally, that they could withdraw from the study at any time without providing a reason. (Line 222-226)

Reviewer #2: 

Thank you for the opportunity to review this well-written paper. I found it interesting to read. There are a few minor areas where clarity is required.

Thank you reviewer 2 for your useful comments and the interesting reflections. 

Aims & rationale: The aims of the study and rationale for the study are clearly outlined – and the originality of the study and the novel contributions it makes to the existing literature are made clear. Lines 45-47 – it is stated that it is assumed that the approximate 60% resilience/recovery rates found in the research on PTEs apply to partners of those with an advanced stage diagnosis…”since in clinical practice, most of these partners have a resilient outcome”. Is this the anecdotal/subjective experience of the authors? If so, this should be stated for clarity’s sake (otherwise the sentence requires a reference). This is then referred to later in the discussion where the findings of this study do indeed support this assumption.

There is a lot of quantitative research on resilience following all kind of PTEs and every study mentions that people follow a resilient process or have a resilient outcome in 60 to 80%. However, to the best of our knowledge, there are no such quantitative studies on resilience in cancer caregivers. Nevertheless, three of the authors are practicing GPs. They can affirm that most of the cancer caregivers indeed have a resilient outcome. 

We made this clearer as follows by replacing the sentence by: 

To the best of our knowledge, there are no quantitative studies available on the prevalence of resilience in family caregivers of loved ones diagnosed with advanced cancer. Nevertheless, based on the authors’ clinical family practice experience, it is reasonable to assume that most partners of patients diagnosed with advanced cancer follow a resilience trajectory and will have a resilient outcome. (Line 47-51)

Methods: The methodological approach used in the study is clearly explained – and sufficient measures are taken to ensure consistency in analysis between researchers, which enhances the reliability and validity of the study. Clarity is required with regards to participant completion of the Dutch Mental Health Continuum-Short Form (MHC-SF): lines 104-106 do not appear to match what is stated in lines 552-553. This needs to be consistent.

Indeed, you are right. Thank you for mentioning this error. There was indeed an inconsistency. Fifteen out of seventeen participants returned the questionnaire at least twice. The results of the questionnaires are depicted in the graphs in supplement 4.

We made this consistent. (Line 126 and Line 733)

Findings and Discussion: The findings are presented logically and suitably evidenced. I found the arguments made convincing. The discussion locates the findings clearly in the existing literature.

Thank you for this nice comment.

*This does not need to be addressed in the paper* (as it is addressed in the implications section – where further research on ‘personal characteristics’ is recommended), however, for interest’s sake, what I found fascinating is that results line up with the understandings offered by psychoanalytic and attachment theories too. The ‘participant characteristics’ referred to as positivity, flexibility and inner strength, could be understood from related theoretical orientations to represent participants’ mental health, personality structures, ego strength, emotion regulation capacities and interpersonal relationship skills, which are influenced by their personal histories of previous loss/es. Current findings suggest that across cultures approximately 60% of the population

---

## [Editor Report · Decision Letter 1]

6 May 2024

Do all roads lead to Rome? An ideal-type study on trajectories of resilience in advanced cancer caregiving.

PONE-D-24-01787R1

Dear Dr. Opsomer,

We’re pleased to inform you that your manuscript has been judged scientifically suitable for publication and will be formally accepted for publication once it meets all outstanding technical requirements.

Kind regards,

Maria Berghs, PhD

Academic Editor

PLOS ONE
---

## [Editor Report · Acceptance letter]

9 May 2024

PONE-D-24-01787R1 

PLOS ONE

Dear Dr. Opsomer, 

I'm pleased to inform you that your manuscript has been deemed suitable for publication in PLOS ONE. Congratulations! Your manuscript is now being handed over to our production team.

Kind regards, 

on behalf of

Dr. Maria Berghs 

Academic Editor

PLOS ONE